# TOM40 Inhibits Ovarian Cancer Cell Growth by Modulating Mitochondrial Function Including Intracellular ATP and ROS Levels

**DOI:** 10.3390/cancers12051329

**Published:** 2020-05-22

**Authors:** Wookyeom Yang, Ha-Yeon Shin, Hanbyoul Cho, Joon-Yong Chung, Eun-ju Lee, Jae-Hoon Kim, Eun-Suk Kang

**Affiliations:** 1Department of Obstetrics and Gynecology, Gangnam Severance Hospital, Yonsei University College of Medicine, Seoul 06273, Korea; wookyeom78@gmail.com (W.Y.); hayeon37@yuhs.ac (H.-Y.S.); hanbyoul@yuhs.ac (H.C.); eunju13@yuhs.ac (E.-j.L.); 2Laboratory of Pathology, National Cancer Institute, National Institutes of Health, Bethesda, MD 20892, USA; chungjo@mail.nih.gov; 3Department of Laboratory Medicine and Genetics, Samsung Medical Center, Sungkyunkwan University School of Medicine, Seoul 06351, Korea

**Keywords:** TOM40, epithelial ovarian cancer, mitochondria, metformin

## Abstract

TOM40 is a channel-forming subunit of translocase, which is essential for the movement of proteins into the mitochondria. We found that TOM40 was highly expressed in epithelial ovarian cancer (EOC) cells at both the transcriptional and translational levels; its expression increased significantly during the transformation from normal ovarian epithelial cells to EOC (*p* < 0.001), and TOM40 expression negatively correlated with disease-free survival (Hazard ratio = 1.79, 95% Confidence inerval 1.16–2.78, *p* = 0.009). TOM40 knockdown decreased proliferation in several EOC cell lines and reduced tumor burden in an in vivo xenograft mouse model. TOM40 expression positively correlated with intracellular adenosine triphosphate (ATP) levels. The low ATP and high reactive oxygen species (ROS) levels increased the activity of AMP-activated protein kinase (AMPK) in TOM40 knockdown EOC cells. However, AMPK activity did not correlate with declined cell growth in TOM40 knockdown EOC cells. We found that metformin, first-line therapy for type 2 diabetes, effectively inhibited the growth of EOC cell lines in an AMPK-independent manner by inhibiting mitochondria complex I. In conclusion, TOM40 positively correlated with mitochondrial activities, and its association enhances the proliferation of ovarian cancer. Also, metformin is an effective therapeutic option in TOM40 overexpressed ovarian cancer than normal ovarian epithelium.

## 1. Introduction

Mitochondria are intracellular organelles that play important roles in various cellular metabolic processes such as ATP production, iron homeostasis, small-molecule biosynthesis, and cellular redox status maintenance [1]. Of the 1000 to 1500 known mitochondrial proteins, 36 are encoded in the mitochondrial DNA. The remaining proteins translocate to the mitochondria after synthesis in the cytosol and are encoded by the nuclear DNA [2,3]. Each mitochondrion consists of a single outer membrane and an inner membrane, which form an aqueous intermembrane space and matrix [4]. Outer membrane protein channels, such as the translocase of the outer membrane (TOM) complex, determine whether mitochondrial proteins localize to the interior of the mitochondria or the mitochondrial membrane [4,5,6].

The TOM complex forms a protein channel in the outer mitochondrial membrane. TOM40 forms the main channel; TOM70, TOM22, and TOM20 are substrate receptors; and TOM7, TOM6, and TOM5 are small accessory proteins [7,8]. The TOM complex is the main conduit, through which most mitochondrial pre-proteins pass, making it very important for maintaining mitochondrial function [6,8]. TOM40, located in the center of the TOM complex, is a channel-forming, chaperone-like protein [9] that helps mitochondrial pre-proteins move through the channel. 

TOM40 has been reported to be associated with late-onset neurodegenerative diseases such as Alzheimer’s disease and Parkinson’s disease [7]. Prior studies demonstrated that loss or knockdown of TOM40 disrupts mitochondrial membrane potential, interferes with the uptake of mitochondria-targeted proteins, and induces mitochondrial stress responses [10,11]. One study showed that decreased TOMM-40 expression in *Caenorhabditis elegans* inhibited growth in the 1st and 3rd larval stages [10]. Another study found that *Tom40* homozygous knockdown mice died during embryonic stage E1 while *Tom40*+/− mice showed normal embryonic development; however, the 2-year survival rate of *Tom40*+/− mice was 30% lower than that of normal animals [12]. In a transgenic mouse experiment, overexpression of TOM40 via lentiviral-mediated delivery improved mitochondrial function by reducing cellular oxidative damage in the hippocampal and cortical regions of the brain and by decreasing inflammation and neurodegeneration [13]. These reports suggest that TOM40 is important for maintaining intracellular mitochondrial function and for preventing cell death, but the correlation between TOM40 and cancer has not yet been examined.

Mitochondria have an essential role in cancer cell energy metabolism. The mitochondrial function in cancer cells alters to accommodate the cellular and environmental conditions around cancer cells in cancer growth, tumorigenesis, and cancer developmental process [14]. Many studies have endeavored to diminish the tumorigenic function of mitochondria using U.S. Food and Drug Administration (FDA)-approved drugs to suppress cancer growth such as analgesics, antibacterial, antipsychotics, antidepressants, antidiabetics, and anticancer drugs [15]. In particular, the anticancer effects of metformin, the first-line therapy for patients with type 2 diabetes, have been demonstrated in both preclinical and clinical studies [16]. Metformin activates AMP-activated protein kinase (AMPK) in cancer cells; inhibits Phosphoinositide 3-kinase/Protein kinase B/mammalian Target of Rapamycin (PI3K/Akt/mTOR) pathway signaling, diminishes insulin and insulin-like growth factor production; and has antimitotic, anti-inflammatory, and antiangiogenic effects [17]. Several studies have reported that metformin inhibits the growth and metastasis of ovarian cancer through AMPK-dependent and AMPK-independent molecular mechanisms [18,19,20,21,22]. In addition, combined administration of metformin and Poly (ADP-ribose) polymerase (PARP) inhibitors or cisplatin-based chemotherapy has been explored in ovarian cancer treatment [23,24]. Moreover, metformin is cytotoxic to ovarian cancer cells but not to normal cells [19].

Most of all, metformin is also known to inhibit mitochondrial complex I. Metformin impaired oxidative phosphorylation (OXPHOS) capacity and mitochondrial respiration and induced lactate production in human platelets [25,26]. Breast cancer cells exposed to metformin showed a reduction in mitochondrial respiration and impaired citric acid cycle activity by lactate overproduction [27]. The anticancer effects of metformin through various molecular signals is highly crucial, but in this study, metformin’s inhibition of the mitochondrial function is further evaluated.

In this study, we investigated the significance of TOM40 overexpression in ovarian cancer, examined the effect of TOM40 on mitochondrial energy metabolism and the growth of ovarian cancer cell lines and xenografts, as well as evaluated the effects of metformin in TOM40-overexpressed ovarian cancer cell lines.

## 2. Results

### 2.1. TOM40 Overexpression Is Negatively Correlated with Disease-Free Survival in Ovarian Cancer Patients

To evaluate the relative expression of TOM40 in epithelial ovarian cancer (EOC) cells and normal ovarian epithelial cells, we used real-time polymerase chain reaction (PCR) to evaluate *TOM40* mRNA levels in four immortalized human ovarian surface epithelial (iHOSE) cell lines and fourteen EOC cell lines. *TOM40* mRNA levels were significantly higher in EOC cell lines than in iHOSE cells, (5.36-fold, *p* = 0.0207) (Figure 1A). In addition, the TOM40 protein expression significantly increased in EOC cells compared to iHOSE cells when normalized to α-actinin (4.12-fold, *p* = 0.0173) (Figure 1B). Furthermore, microarray results from our previous reports showed that the expression of *TOM40* increased 5.55-fold in YDOV-139 and 4.06-fold in YDOV-157 cell lines, compared to iHOSE cells [28,29,30]. In addition, three datasets from the Gene Expression Omnibus (GEO) databased were analyzed, which compared gene expression profiles of EOC with iHOSE or LMP (Low malignant potential) tissues (GEO accession nos. GSE18520, GSE26712, and GSE9899). Expression of *TOM40* significantly increased in EOC tissues compared to iHOSE or LMP tissues in all three datasets (1.59-fold, Cancer/iHOSE in GSE18520; 1.83-fold, Cancer/iHOSE in GSE26712; and 1.33-fold, Cancer/LMP in GSE9899; *** *p* < 0.001, *** *p* < 0.001, and ** *p* < 0.01, respectively) (Figure 1C). To determine whether TOM40 expression is linked to clinicopathological features of EOC, we performed immunohistochemistry in normal, benign, borderline, and EOC tissues derived from patients. TOM40 expression was observed in the cytoplasm of malignant and normal cells (Figure 1D). TOM40 expression increased according to tumorigenic progression status (benign = 1.37-fold, borderline = 2.15-fold, and EOC = 2.82-fold compared to normal, *** *p* < 0.001) (Figure 1E). Relative TOM40 expression levels by clinicopathologic characteristics of ovarian cancer patients are summarized in Table 1. TOM40 expression was greater in type II tumors that include high-grade serous carcinoma and undifferentiated tumors (histoscore = 241, *n* = 114) than in type I tumors that include endometrioid, clear cell, mucinous, and transitional tumors (histoscore = 219, *n* = 81) (*p* = 0.005) (Table 1). We next examined the relationship between TOM40 expression and clinical outcomes in 181 EOC patients. Of 181 EOC tumors, 91 (50.3%) overexpressed TOM40. TOM40 overexpression significantly correlated with worse disease-free survival (*p* = 0.027) (Figure 1F), and it was associated with worse overall survival (*p* = 0.328) (Figure 1G). Patients with advanced International Federation of Gynecology and Obstetrics (FIGO) stage tumors, serous type tumors, and poor tumor grades showed significantly worse disease-free survival (*p* < 0.001, *p* < 0.001, and *p* = 0.002, respectively) and overall survival (*p* = 0.001, *p* = 0.002, and *p* = 0.0142, respectively) than patients with early FIGO stage, non-serous type, and good/fair tumor grades (Appendix A). Univariate and multivariate analyses for all clinicopathological characteristics and survival are shown in Table 2. The disease-free survival rate was 39.6% for patients with TOM40 overexpression compared with 54.4% for patients with weak/negative expression (hazard ratio (HR) = 1.57, 95% CI, 1.04–2.36 in univariate analysis; HR = 1.73, 95% CI, 1.12–2.67 in multivariate analysis), whereas it was not associated with overall survival (Table 2). These results indicate that TOM40 is overexpressed in EOC compared to normal epithelial ovarian cells, and patients with EOC tumors that express TOM40 at high levels have worse prognoses than patients with EOC tumors that express low levels of TOM40.

### 2.2. TOM40 Knockdown Suppresses the Growth of EOC Cell Lines and Xenografts

To determine the effect of decreased TOM40 expression in EOC cell lines, we used a lentiviral system. TOM40 knockdown EOC cells (sh-TOM40) were generated using a pLKO.1 sh-TOM40 vector, and sh-control cells were made using a pLKO.1 empty vector. To confirm TOM40 knockdown, we performed western blot analysis. TOM40 expression was markedly lower in sh-TOM40 cells compared to sh-control cells (Figure 2A, representative images, right). To assess the potential impact of TOM40 expression on proliferation, we seeded 1 × 10^6^ EOC cells that expressed either sh-control or sh-TOM40. After 48 h, the number of cells was directly counted using an automated cell counter. There were significantly fewer sh-TOM40 EOC cells compared to sh-control EOC cells (Figure 2A). To measure cell growth rate over four days, we used a crystal violet assay. We found that the growth rate of sh-TOM40 EOC cells was significantly slower than the sh-control cells (Figure 2B). Comparing sh-TOM40 cells to sh-control cells, the proliferative rates of sh-TOM40 cells were 66.08%, 53.79%, 75.39%, 53.33%, and 89.37% for RMUG-S, TOV-112D, SK-OV-3, OVCA-429, and OVCA-433 cells, respectively after 4 days (Figure 2B). To evaluate the long-term impact of TOM40 expression on proliferation, we performed a colony forming assay with EOC cells that stably expressed sh-TOM40 or sh-control. The size of sh-TOM40 EOC colonies was visibly smaller than sh-control cell colonies (Figure 2C, representative images, top). The growth rates of sh-TOM40 cells compared to sh-control cells were 67.28%, 23.96%, 62.92%, 19.13%, and 37.58% in RMUG-S, TOV-112D, SK-OV-3, OVCA-429, and OVCA-433, respectively, after 10 day (Figure 2C, bar graph). To determine the effects of TOM40 knockdown in vivo, we injected 3 × 10^6^ RMUG-S cells, which stably expressed either sh-control or sh-TOM40, subcutaneously into a BALB/c nude mice to induce growth of xenograft tumors. Tumors were measured from day 14 through day 68 post-injection. Tumors harvested 68 days post-injection are displayed in Figure 2D. Comparing the sh-TOM40 RMUG-S-derived tumors to the sh-control RMUG-S-derived tumors, the volume of the sh-TOM40 tumors was 51.48% of the sh-control tumors 68 days post-injection; *** *p* < 0.001 (Figure 2E). Furthermore, the final volume and weight of the sh-TOM40 RMUG-S-derived tumors were significantly less than that of the sh-control tumors (Figure 2F). Additionally, to evaluate the effects of TOM40 overexpression, we engineered SK-OV-3, OVCAR-3, and iHOSE-8695-SV40 cells that stably expressed high levels of TOM40 or endogenous TOM40 levels (empty plasmid) using a lentiviral system. When we attempted to express TOM40 exogenously in two ovarian cancer cells, exogenous TOM40 protein faintly overexpressed more than iHOSE-8695-SV40 (Appendix A, Western panel). We evaluated the proliferation of these cells for seven days using a crystal violet assay. The growth rate of the TOM40-overexpressing SK-OV-3 and iHOSE-8695-SV40 cells was significantly faster than the control cells (empty); conversely, the OVCAR-3 control cells (empty) grew faster than the OVCAR-3-TOM40 cells (Appendix A). These results indicate that TOM40 knockdown significantly reduced the growth of EOC cell lines; however, the effects of TOM40 overexpression on proliferation may be cell line specific. 

### 2.3. TOM40 Expression Is Positively Correlated with Intracellular ATP Levels

Intracellular ATP levels were quantified in 14 EOC and 2 iHOSE cell lines luciferase-based assay. We found that intracellular ATP levels were significantly higher in EOC cell lines than in iHOSE cell lines (2.03-fold, *** *p* < 0.001) (Figure 3A). In addition, TOM40 knockdown (sh-TOM40) significantly reduced intracellular ATP levels by an average of 12.69% ± 0.01% (*** *p* < 0.001) compared to sh-control EOC cells (Figure 3B). To determine whether changes in intracellular ATP levels resulting from TOM40 knockdown were due to a change in the quantity of mitochondria, we compared the number of mitochondria in cell lines stably expressing sh-TOM40 or sh-control using real-time PCR. The relative number of mitochondria was calculated as the ratio of mitochondrial 16S RNA to genomic β2-microglobulin. We found that TOM40 knockdown-induced changes in intracellular ATP levels were independent of changes in mitochondrial numbers (Figure 3C). We then evaluated intracellular ATP levels in EOC cell lines and iHOSE cell lines that stably overexpressed TOM40 or expressed endogenous levels of TOM40 (empty). Intracellular ATP levels increased by 13.1% ± 0.02% in TOM40-overexpressing EOC cells but, in TOM40-overexpressing iHOSE cells, significantly decreased compared to controls (empty) (Figure 3D). There was an increase in the number of mitochondria in the TOM40-overexpressing EOC cells but not in the corresponding iHOSE cells (Figure 3E). These results suggest that the rise in intracellular ATP levels in TOM40-overexpressing EOC cells may be due to an increase in the number of mitochondria.

To determine whether mitochondrial oxidative phosphorylation (OxPhos) activity is associated with declined intracellular ATP levels in TOM40 knockdown EOC cells, mitochondrial OxPhos activity was indirectly measured using WST-8/crystal violet assay (for dehydrogenase enzyme activity) and cytochrome c oxidase activity assay (for complex IV activity). Dehydrogenase activity and cytochrome oxidase activity were not correlated with intracellular ATP levels in sh-TOM40 stably expressed EOC cells since the results of enzyme activity were not consistent among all cells (Appendix A). To confirm the expression of OxPhos enzymes in TOM40 knockdown cells, we performed western blot analysis using an anti-OxPhos cocktail antibody. The expression of COX II-22kDa protein in mitochondrial complex IV was slightly altered in response to TOM40 expression in TOV-112D, SK-OV-3, and OVCAR-3 cells (Appendix A). However, almost all OxPhos enzyme expression was not significantly altered in EOC cells. These results may suggest that declined intracellular ATP levels in TOM40 knockdown EOC cells do not correlate with mitochondrial OxPhos activity.

### 2.4. TOM40 Expression Knockdown in EOC Cells Increases Mitochondrial Membrane Potential and Intercellular Reactive Oxygen Species

To determine the effect of TOM40 expression on mitochondrial membrane potential, we measured mitochondrial membrane potential in four EOC cell lines that stably expressed sh-control or sh-TOM40, using tetraethylbenzimidazolylcarbocyanine iodide (JC-1) staining, tetramethylrhodamine ethyl ester (TMRE) staining, fluorescence imaging, and Fluorescence-activated cell sorting (FACS) analysis. We found that the intensity of JC-1 red-fluorescence increased in sh-TOM40 stably expressed EOC cells comparing with sh-control cells (Increasing rate: 11.52%, 11.19%, 35.76%, and 13.17% in RMUG-S, TOV-112D, OVCA-429, and OVCAR-3 cell, respectively, **** *p* < 0.0001) (Figure 4A, the upper region in the dot plot). Representative fluorescence images of JC-1 staining in four EOC cells is shown in Figure 4B. Although JC-1 was well stained in EOC cells, it was difficult to distinguish the difference of fluorescence intensity in all images. EOC cells were co-stained with MitoTracker Green FM (independent mitochondrial membrane potential) and TMRE. TOM40 knockdown enhanced Mitotracker Green FM staining in TOV-112D and OVCA-429 but insignificantly deceased in OVCAR-3 cells (Figure 4C, green histograms and bar graph). TMRE staining showed that mitochondrial membrane potential was higher in RMUG-S, TOV-112D, OVCA-429, and OVCAR-3 cells that stably expressed sh-TOM40 compared to sh-control (Increasing rate: 9.86%, 13.39%, 10.95%, and 24.27% in RMUG-S, TOV-112D, OVCA-429, and OVCAR-3 cells, respectively, *** *p* < 0.001) (Figure 4C, red histograms and bar graph). Representative fluorescence images of TMRE and Hochest 33342 staining in OVCA-433, RMUG-S, and SK-OV-3 cells are shown in Figure 4D. To determine the levels of reactive oxygen species (ROS) by TOM40 knockdown in EOC cells, ROS was measured by FACS analysis using 2’,7’-dichlorofluorescin diacetate (DCFDA) and CellROX-Green dye staining. Although there is a difference in the rate of increased ROS levels in each cell, TOM40 knockdown robustly induced intracellular ROS production in EOC cell lines (Figure 5A and Appendix A). Moreover, we found that declined ROS levels by using N-acetyl-L-cysteine (NAC) and glutathione (GSH) increased cell growth rate in sh-TOM40 stably expressed EOC cells (Figure 5B). These results imply that changes in TOM40 expression affect the mitochondrial capacity to produce ATP by altering mitochondrial membrane potential and ROS production.

### 2.5. Metformin Inhibits the Proliferation of TOM40-Overexpressing EOC Cells in an AMPK-Independent Manner

When TOM40 expression was reduced in EOC cells, ATP levels decreased while ROS levels increased (Figure 4B and Figure 5 respectively). Both reduction in ATP and increase in ROS are well known to activate AMP-activated protein kinase (AMPK), an important regulator of cellular metabolism and energy homeostasis [31,32]. To determine whether AMPK is activated by TOM40 knockdown in EOC cell lines, western blotting was performed. We found that phosphorylation of AMPK increased in TOM40 knockdown cell lines compared to sh-control cell lines and that the phosphorylation of ACC, a downstream target of AMPK, also increased (Figure 6A). Interestingly, when OVCA-429 cells were exposed to glucose deprivation, the activation of AMPK was observed in sh-TOM40-expressed cells but was not less activated in TOM40-overexpressed cells at both normal glucose and glucose-free conditions (Appendix A). To investigate whether the activation of AMPK attenuated TOM40 knockdown EOC cell growth, sh-TOM40 stably expressed EOC cells were transfected with si-AMPKα1. We found that AMPK knockdown in sh-TOM40 expressed EOC cells (Figure 6B, right penal) did not influence growth suppression in either RMUG-S, TOV-112D, or OVCAR-3 cell lines (Figure 6B, Left bar graph). To investigate whether the 5-aminoimidazole-4-carboxamide ribonucleotide (AICAR), metformin, or aspirin, well-known as AMPK activators, attenuated EOC cell growth, EOC cells were treated with 1 mM AICAR, 5 mM metformin, and 1 mM aspirin. We found that metformin significantly inhibited proliferation in OVCAR-3 and SK-OV-3 cell lines, including those that overexpressed TOM40, and that the effect was enhanced during glucose deprivation (Figure 6C,D). However, AICAR and aspirin did not or insignificantly inhibited EOC cell growth (Figure 6C). Although the phosphorylation of AMPK is significantly elevated in the presence of metformin and AICAR under normal glucose and glucose-free conditions (Figure 6E), the current results indicated that the association between TOM40 expression and AMPK activation is too weak in EOC cell growth. Besides, cell growth inhibition of metformin was not caused by apoptosis due to the absence of cleaved PARP and caspase-3 (Figure 6E). These results indicate that metformin can significantly inhibit EOC cell growth in an AMPK-independent manner but does not induce apoptosis.

## 3. Discussion

Normal cells produce energy through mitochondrial oxidative phosphorylation (OxPhos); however, cancer cells, due to mitochondrial defects, produce most of their energy by anaerobic glycolysis. This is known as the Warburg effect [33,34]. Therefore, due to their inefficient energy metabolism, cancer cells require large amounts of glucose to sustain proliferation [35,36]. Historically, it has been assumed that glycolytic metabolism (vs. mitochondrial metabolism) is essential for cancer cell proliferation; however, not all cancer cells attain energy via the Warburg effect. Furthermore, prior studies have reported that cancer cells can produce copious amounts of ATP through OxPhos and that inhibition of OxPhos induces cell cycle arrest [36,37]. These findings suggest that not all cancer cell mitochondria are defective and that they may play an important role in cancer cell proliferation. Therefore, it seemed that cancer cell requires the mitochondria for energy production. TOM40 is closely associated with mitochondrial function and activity in cancer cells [38,39]. TOM40, an outer membrane channel protein essential for maintaining mitochondrial function, modulates the translocation of pre-mitochondrial proteins into the mitochondria [6,7,8], and when TOM40 expression increases, overall mitochondrial function and activity increases [13,40]. Since mitochondria play a critical role in promoting cancer cell growth and TOM40 enhances the mitochondrial activity, reduction of TOM40 expression is anticipated to suppress mitochondrial function and to inhibit cancer growth. We found that TOM40 is closely associated with the mitochondria in cancer cells and modulates ATP production and cancer cell growth. TOM40 was more highly expressed in EOC cell lines (Figure 1A,B), GEO datasets (Figure 1C), and cancer tissue microarrays (Figure 1D–G and Table 1) compared to normal ovarian epithelial cells. Moreover, knockdown of TOM40 suppressed EOC proliferation (Figure 2). Also, intracellular ATP levels were higher in ovarian cancer cells than in iHOSE cells (Figure 3A). TOM40 knockdown decreased intracellular ATP levels (Figure 3B); conversely, TOM40 overexpression induced ATP levels in ovarian cancer cells (Figure 3D). However, the altered ATP levels were not correlated with mitochondrial number (Figure 3C,E). To determine a reason behind TOM40 knockdown modulating intracellular ATP levels, we conducted OxPhos activity assay and performed western blot analysis to determine the expression levels of OxPhos related proteins. We could not conclude a relationship between OxPhos and TOM40 expression since the results in EOC cells were not consistent (Appendix A). Therefore, TOM40 reduction could not be correlated with the alteration of OxPhos activity in the reduction of ATP levels. Instead, we found that TOM40 knockdown consistently induced mitochondrial membrane potential and intracellular ROS levels (Figure 4 and Figure 5). In previous reports, when the mitochondrial membrane potential is out of the normal range (dropped or enhanced) for a long duration, that decreased cell viability and impaired mitochondria [41]. High mitochondrial membrane potential produces significantly high levels of intracellular ROS [42,43,44]. In turn, excessive ROS generation causes mitochondrial and cellular damage [41,44]. Hence, changes in the intracellular ATP levels following TOM40 knockdown can be caused by excessive ROS generation through the increased mitochondrial membrane potential.

We attempted to determine the causes for the alteration of membrane potential along with ROS and ATP levels by TOM40 expression. A previous report suggested that overexpressing TOM40 increased TOM20 expression as a subunit of the TOM complex [38]. To reveal whether the cellular physiologic effects following TOM40 knockdown are caused by the alteration of TOM complex subunits expression, we aimed to detect TOM complex subunits expression levels in EOC cells. Our results showed that TOM40 knockdown in EOC cells reduced levels of TOM complex subunits and some mitochondrial proteins, including TOM22, TOM20, and SDHA (Appendix A). These reductions of TOM22 and TOM20 expression did not modulate transcriptional levels (Appendix A). To characterize the molecular mechanism by which TOM40 knockdown reduces the expression of TOM22 and TOM20, we evaluated the effects of protease, proteasome, and lysosome inhibitors on TOM40 knockdown EOC cells (Appendix A). Although protease inhibitors restored TOM40 knockdown-induced reduction of TOM22 expression in OVCA-429 cells, other EOC cell lines either did not display the same phenomenon or slightly restored expression (Appendix A). Also, lysosome inhibitors did not change TOM complex subunits (Appendix A). Based on this data, TOM40 may modulate the protein levels of TOM22 and TOM20; however, there are cell line-specific or TOM complex subunit-specific effects. Hence, the precise mechanism by which TOM40 affects the expression of other TOM complex proteins has yet to be fully characterized. Furthermore, when TOM40 is knocked down, the transport of mitochondrial pre-proteins is disrupted, which affects mitochondrial membrane potential and induces a mitochondrial stress response [10,11,38]. To assess the effect of TOM40 knockdown on protein precursors trafficking into the mitochondria, we identified the cellular location of TOM complex subunits and mitochondrial proteins by subcellular fractionation. We found that the TOM40 knockdown weakly affected the localization of mitochondrial pre-proteins in EOC cell lines (Appendix A). We hypothesize that TOM40 knockdown may have been insufficient to affect mitochondrial pre-protein trafficking. Unexpectedly, we found substantial TOM20 expression in the nucleus as well as in the mitochondria (Appendix A). This subcellular localization warrants further investigations. Finally, these results suggested that the changes in membrane potential, ROS levels, and ATP levels, according to TOM40 expression, can be caused by more complex mechanisms and that new approaches may be needed to fully elucidate them.

The subcellular changes of declined intracellular ATP and increased ROS levels can activate AMPK, a pivotal player in cellular metabolism and energy homeostasis [45,46]. Overexpressing AMPK subunits are clinically correlated with prognosis of ovarian cancer [47]. Consistent with these reports, we found that TOM40 knockdown activated AMPK in most EOC cell lines that we evaluated (Figure 6A); TOM40-overexpressing OVCA-429 cells showed no AMPK activation under the same conditions and weak AMPK activation even under glucose-free condition in comparison to control cells (Appendix A). However, AMPKα1 knockdown did not influence sh-TOM40 stably expressed EOC cell growth (Figure 6B). AICAR and aspirin, as AMPK activators [48,49], did not suppress EOC cell growth (Figire 6C). Uniquely, metformin induced the suppression of EOC cell growth (Figure 6C,D), not through apoptosis (Figure 6E). Metformin activates AMPK-mediated AMPK/mTOR signaling cascade [16,50], and it has been shown that metformin effectively suppresses ovarian cancer growth in preclinical and clinical studies [50,51,52,53]. However, our findings imply that the effects of metformin in EOC cell growth were not associated with AMPK activation. Notably, high-dose metformin suppressed mitochondrial complex I activity by inhibiting nicotinamide adenine dinucleotide (NADH) oxidation [54]. Previous reports suggested that overexpressed TOM40 increased activities of the oxidative phosphorylation complexes I and IV [38]; also, most ovarian cancer cell lines have higher mitochondrial activity than iHOSE cell lines [55]. Therefore, we assumed that metformin significantly reduced EOC proliferation in an AMPK-independent manner in endogenous or exogenous TOM40 overexpressed EOC cell lines.

## 4. Materials and Methods

### 4.1. Patients, Tumor Samples, and Tissue Microarrays

A total of 336 patients diagnosed with ovarian tumors (84 benign, 51 borderline, and 201 cancerous) at Gangnam Severance Hospital (Seoul, Republic of Korea) were enrolled in this study between 1993 and 2014. Some of the paraffin blocks were provided by the Korea Gynecologic Cancer Bank through the Bio & Medical Technology Development Program of the Ministry of the National Research Foundation (NRF) funded by the Korean government (MSIT) (NRF-2017M3A9B8069610). Before beginning the study, approval was obtained from the Institutional Review Board of Gangnam Severance Hospital (IRB No. 3-2010-0030 and 3-2011-0057), and written consent to participate in the study was obtained from each person. Detailed immunohistochemistry (IHC) protocols were performed as described previously [56]. Briefly, sections were incubated with rabbit polyclonal anti-TOM40 antibody (Cat. 18409-1-AP; Proteintech Group, Inc, Rosemont, IL, USA) at a dilution of 1:150, overnight at 4 °C. Negative controls omitted the primary antibody. Human testicular tissue was used as a positive control for TOM40 immunoreactivity. The stained sections were digitized utilizing the NanoZoomer 2.0 HT (Hamamatsu Photonics K.K., Hamamatzu, Japan) at ×20 objective magnification (0.5 μm resolution). For tissue microarrays, digital analysis of TOM40 IHC was performed using Visiopharm software v4.5.1.324 (Visiopharm, Hørsholm, Denmark). 3,3’-diaminobenzidine (DAB) (brown staining) intensity (0 = negative, 1 = weak, 2 = moderate, and 3 = strong) was obtained using a predefined algorithm and optimized settings. The overall histoscore was calculated as the percentage of positive cells multiplied by their staining intensity (possible range 0–300) [57]. The cutoff values of histoscore were defined by considering the distribution and prognostic significance of the values.

### 4.2. Cell Culture and Antibodies

Ovarian cancer (EOC) cell lines and immortalized human ovarian surface epithelial (HOSE) cells were obtained and cultured as described previously [56,58]. All purchased cell lines were maintained in culture media supplemented with 10% Fetal bovine serum (FBS), (Sigma-Aldrich, St. Louis, MO, USA) and 1% penicillin/streptomycin (Mediatech, Inc., Manassas, VA, USA). Anti-TOM40 (18409-1-AP) antibodies were purchased from Proteintech Group, Inc (Chicago, IL, USA). Anti-TOM70 (sc-390545), anti-TOM22 (sc-58308), anti-TOM40 (sc-365467), anti-α-actinin (sc-17829), and anti-LaminB (sc-6216) were obtained from Santa Cruz Biotechnology (Santa Cruz, CA, USA). Mitochondrial marker antibody sampler kit (#8674, which included anti-SDHA, anti-VDAC, anti-pyruvate dehydrogenase, and anti-COXIV), anti-PHB1 (#2426), anti-HSP60 (#12165), phospho-AMPKα (Thr172, #2535), phospho-ACC (Ser79, #3661), anti-ACC (#3676), phospho-LDHA (Tyr10, #8176), anti-LDHA (#3582), anti-Bim (#2819), anti-Bax (#3582), anti-Bad (#9239), anti-Bcl-2 (#2870), anti-cleaved PARP (#5625), anti-cleaved caspase-3 (#9664), anti-cytochrome c (#4280), anti-SOD1 (#4266), and anti-TOM20 (#13929) were purchased from Cell Signaling Technology (Danvers, MA, USA). Anti-AMPKα1 (cat. AF3197) was obtained from R&D Systems (Minneapolis, MN, USA). Total OXPHOS Human WB Antibody Cocktail (cat. ab110411) was purchased from Abcam (Cambridge, United Kingdom).

### 4.3. SYBR Green Real-Time PCR

RNA extraction, cDNA synthesis, and the SYBR Green real-time PCR protocol were performed as described previously [56]. The primers for PCR analysis were as follows: TOM40 forward (5’-GGCCCCGGTCTCAGGTCCAA-3’), TOM40 reverse (5’-CCAGGGTGACGGCTGCTGTG-3’), TOM70 forward (5’-CAAAGCATGCTGTTAGCCGA-3’), TOM70 reverse (5’-TCCTTTAAGCATGGGCTGGG-3’), TOM22 forward (5’-GCCGGAGCCACTTTTGATCT-3’), TOM22 reverse (5’-GAGAGCCCTGTGTTAGGTCC-3’), TOM20 forward (5’-GAGAGAGCTGGGCTTTCCAA-3’), TOM20 reverse (5’-TGTCAGATGGTCTACGCCCT-3’), TOM7 forward (5’-GGTTGCTGTAAGGGGTCCTC-3’), TOM7 reverse (5’-GTTCAGGCATTCCGGGATCT-3’), TOM6 forward (5’-CCGCTTTGCCACTGATAGGA-3’), TOM6 reverse (5’-GGGTTCAGCACAGGATTCCA-3’), TOM5 forward (5’-CTCCTGCGAGTCACTCCATT-3’), TOM5 reverse (5’-GCCTATTCACTTGCAGAGAGG-3’), β-actin forward (5’-ATTAAGGAGAAGCTGTGCTACGTC-3’), and β-actin reverse (5’-ATGATGGAGTTGAAGGTAGTTTCG-3’). β -actin was used to normalize the quantity of cDNA used for PCR. Relative messenger RNA expression was quantified using the comparative Ct (△Ct) method and expressed as 2^−△△Ct^, where △△Ct = △E − △C, △E = Ct_E target_ – Ct_E β-actin_, and △C = Ct_c target_ – Ct_c β-actin_ (E, experimental result; C, control).

### 4.4. Gene Expression Omnibus Dataset Analysis

Gene expression profiling data were extracted from the GSE18520, GSE26712, and GSE9899 microarray datasets from Gene Expression Omnibus (GEO). Identification of differentially expressed TOM40 was conducted with a sorting tool based on Microsoft Excel software (Probe number 202264_s_at, Affymetrix Human U133A Platform, gene accession number NM_006144.). Box plots and statistical analyses were performed using GraphPad Prism 7 software (GraphPad Software, Inc., La Jolla, CA, USA).

### 4.5. Engineering Cell Lines That Stably Express TOM40 or sh-TOM40

To generate cell lines that stably express sh-TOM40, pLKO.1, and TOM40, shRNA libraries were purchased from Open Biosystems (Waltham, MA, USA). Of the five lentiviral constructs tested, we used the one with the best knockdown efficiency in our experiments. It incorporated the following human TOM40 shRNA sequence: 5’-AAACCCAGGGCTGCCTTGGAAAAG-3’. The pLKO.1 empty vector was used to create sh-control cells. To establish TOM40-overexpressing cells, we cloned TOM40 from the pCDH-CMV-TOM40-EF1-copGFP+ puro expression vector with the following PCR primers: TOM40 forward (5’-TTCTAGAGCCACCATGGGGAACGTGTTGG-3’) and TOM40 reverse (5’-TGCGGCCGCTCAGCCGATGGTGAGGCCAAAG-3’). The pCDH empty vector was used to make control cells for the TOM40-overexpressing cells. Stably expressing cell lines were generated using the viral packaging plasmids, pMD2.G and psPAX2. Virus particles were collected 48 hours and 72 hours post-transfection and used to infect the cells. Infected cells were selected with 1 μg/mL puromycin (Sigma-Aldrich) for 15 days.

### 4.6. Proliferation and Colony Formation Assays

To assess proliferation by directly count cell numbers, cells were seeded into the wells of a 6-well plate at a density of 1 × 10^6^ cells and cultured for 48 hours. Then, the cells were trypsinized and resuspended in 100 μL complete media. The number of cells in 10 μL of the cell suspension was calculated using a LUNAII^TM^ automated cell counter (Logos Biosystems, Inc., Anyang-si, Korea). To measure cell growth rate, the cells were seeded in a 24-well plate at a density of 1 × 10^5^ cells and were cultured for indicated periods up to 4 days. Cells were stained with a 0.5% crystal violet solution and extracted using 2% Sodium dodecyl sulfate (SDS) solution. To evaluate colony forming assay, the cells were seeded in 35 mm dish at a density of 1 × 10^4^ cells and were grown for up to 11 days. Cells were fixed using 10% acetic acid solution with 10% methanol, stained with 0.5% crystal violet for 15 min, and photographed. crystal violet was extracted using a 2% SDS solution. The crystal violet extract was measured at an absorbance of 595 nm with VERSA max^TM^ (Molecular Devices, Sunnyvale, CA, USA). All experiments were performed in triplicate.

### 4.7. Xenograft Mouse Model of Ovarian Cancer

Five-week old female BALB/c nude mice were injected in the left and right flanks with 3 × 10^6^ RMUG-S ovarian cancer cells that stably express sh-TOM40 group or sh-control (*n* = 3 mice per cell type). Tumor dimensions were measured with digital calipers at indicated periods, and tumor volume was calculated according to the following formula: tumor volume (mm^3^) = length × width^2^ × 0.5. Tumor xenografts were harvested 68 days post-injection. All animal experiments were approved by the Institutional Animal Care and Use Committee at Yonsei University in The Republic of Korea.

### 4.8. Measurement of Intracellular ATP Levels

Intracellular ATP levels were measured using the ApoSENSOR^TM^ ATP Cell Viability Bioluminescence Assay Kit (BioVision, Inc., Milpitas, CA, USA) according to the manufacturer’s instructions. Briefly, the cells were seeded in 10 cm culture dishes at 70% cell density. The cells were treated with FX11, glucose, or glutamine and were incubated for 24 h or 48 h. Cells were cultured in glucose- or glutamine-free DMEM (Gibco, NY, USA cat. 11966-025 and cat. A14430-01, respectively) or glucose-free Rosewell Park Memorial Institute (RPMI) medium 1640 (Gibco, cat. 11879-020). Thereafter, the cells were treated with luciferase assay lysis buffer (Biosesang, Gyeonggi-do, Korea). Cell lysates were sonicated with a VibraCell^TM^ Sonicator (Sonics & Materials, Inc., Newtown, CT, USA) at 30% power for 20 s. The assay mixtures containing 20 µL cell lysate and 100 µL ATP assay buffer were pipetted into the wells of black 96-well plates. Luminescence was measured using the Fluostar Optima (BMG Labtech, Victoria, Australia). A bicinchoninic acid (BCA) assay was performed to quantify protein in cell lysates. Relative intracellular ATP levels were quantified using the ratio of an ATP luminescence value divided by protein levels of each treatment group.

### 4.9. Determination of Mitochondrial DNA Copy Number

Total DNA was extracted from TOM40-overexpressing cells, sh-TOM40 knockdown cells, and control cells using a G-spin Total DNA Extraction Mini Kit (iNtRON Biotechnology, Seoul, Korea) according to the manufacturer’s suggested protocol. The DNA concentration was measured using a NanoDrop 1000 spectrophotometer (Thermo Fisher Scientific, Rockford, IL, USA). SYBR Green real-time PCR was performed using a 10 ng DNA template and primer sets. The primers for the PCR analysis were as follows: mtDNA 16S rRNA forward (5’-GCCTTCCCCCGTAAATGATA-3’), mtDNA 16S rRNA reverse (5’-TTATGCGATTACCGGGCTCT-‘3), β2-microglobulin forward (5’-TGCTGTCTCCATGTTTGATGTATCT-3’), and β2-microglobulin reverse (5’-TCTCTGCTCCCCACCTCTAAGT-3’). β2-microglobulin was used to quantify nuclear DNA, and mtDNA 16S rRNA was used to quantify mitochondrial DNA. Relative mitochondrial DNA copy number was quantified using the comparative Ct (△Ct) method and expressed as 2^−△△Ct^, where △△Ct = △E − △C, △E = Ct_E 16s rRNA_ − Ct_E β2-microglubulin_, and △C = Ct_c 16s rRNA_ − Ct_c β2-microglubulin_ (E, experimental result; C, average Ct of iHOSE cells). 

### 4.10. siRNA Transfection

The cells were seeded at 5 × 10^5^ cells/well in a 6-well plate. Medium without antibiotics was added to each well so that the cells grew to 70% confluence when the transfection was conducted. The 200 pmole siRNA transfection was prepared using Lipofectamine RNAiMAX Reagent (Life Technologies, Rockford, IL, USA) according to the manufacturer’s instructions. si-Luciferase and si-AMPKα1 siRNA were purchased from BIONEER (Cat. SP-3002 and 5562-1, Seoul, Korea).

### 4.11. Mitochondrial Membrane Potential Assay

For JC-1 staining assay (Cat. T3168, ThermoFisher Scientific), the EOC of 2 × 10^5^ cells per tube were stained by 2 μM JC-1 for 30 min at 37 °C CO_2_ incubator. The cells were analyzed with a FACScan flow cytometer (BD Biosciences, San Jose, CA) for quantifying 488 nm excited fluorescence signals at 574 nm (JC-1 aggregate, red) and 520 nm (JC-1 monomer, green). EOC cells were seeded in 6-cm culture dishes at a density of 1 × 10^5^ cells. A mitochondrial membrane potential assay was performed using the Membrane Potential Assay Kit (II) (#13296S, Cell Signaling Technology, Danvers, MA, USA) and MitoTracker Green FM (Thermo Fisher Scientific, Rockford, IL, USA) according to the manufacturer’s instructions. Briefly, cells were stained with 25 nM TMRE and 100 nM MitoTracker Green FM (ThermoFisher Scientific, Rockford, IL, USA) for 15 min in a 5% CO_2_ incubator. Then, the cells were analyzed with a FACScan flow cytometer (BD Biosciences, San Jose, CA). The excitation wavelength is 488 nm, and the emission wavelengths are 578 nm (TMRE, red) and 516 nm (Mitotracker, green). Fluorescence-activated cell sorting (FACS) data were analyzed for dot plots and histogram plots using Flowing Software version 2.5.1 (Turku bioimaging, Turku, Finland). Fluorescent images of the cells were acquired using an EVOS FL cell imaging system (Thermo Fisher Scientific, Rockford, IL, USA).

### 4.12. Measurement of Intracellular Reactive Oxygen Species

To measure intracellular reactive oxygen species using 2’,7’-Dichlorofluorescin Diacetate (DCFDA), sh-control and sh-TOM40 cells were stained with 20 μM DCFDA for 30 min in a 5% CO_2_ incubator. The cells were detached with a trypsin-Ethylenediaminetetraacetic (EDTA) solution and were analyzed with a FACScan flow cytometer (BD Biosciences, San Jose, CA, USA) using excitation and emission wavelengths of 488 nm and 535 nm, respectively. sh-Control and sh-TOM40 cells were stained with 5 μM CellROX Oxidative Stress Reagents (Molecular probes by Life Technology, Carlsbad, CA, USA). Stained cells were incubated for 30 min in a 5% CO_2_ incubator and detached with a trypsin-EDTA solution. CellROX-stained cells were analyzed with a FACScan flow cytometer (BD Biosciences) using excitation and emission wavelengths of 488 nm and 520 nm, respectively. FACS data were analyzed for dot plots and histogram plots using Flowing Software version 2.5.1 (Turku bioimaging, Turku, Finland).

### 4.13. Cytochrome Oxidase Activity Assay

Cytochrome c oxidase activity was measured using the cytochrome oxidase activity colorimetric assay kit (BioVision, Inc., Milpitas, CA, USA) according to the manufacturer’s instructions. Briefly, the cells were harvested in a 10 cm culture dish at 70% cell density. The quantification of protein was measured by Bicinchoninic acid (BCA) assay. The assay mixture containing 20 μL reduced cytochrome C, 100 μL assay buffer, and 150 μg cell lysate was pipetted into the wells of 96 well plates. The kinetic enzyme assay was measured at an absorbance of 550 nm with VERSA maxTM to obtain the variation of colorimetric value. The kinetic protocol is that the interval time is 1 min at 25 °C by 25 min. The rate of the reaction is relative to sh-control samples. The rate was calculated in linear rage. All experiments were performed in duplicate.

### 4.14. Statistical Analysis

The Kruskal–Wallis and Mann–Whitney U tests were used to compare TOM40 expression in the different groups. Survival data, used to generate Kaplan–Meier survival curves, were available for only 181 of the 201 patients with EOC. Differences in survival were analyzed with the log-rank test. Multivariate analyses with hazard ratios (HR) for recurrence were performed using the Cox proportional hazards model. Statistical analyses were performed using SPSS v21.0 (SPSS Inc., Chicago, IL, USA). The statistical analyses of general molecular experiments were performed using Graphpad Prism 7 software. The unpaired t-test was used to compare the control group with experimental group. Results are expressed as the mean ± S.E, with *p* < 0.05 considered statistically significant (^#^
*p* > 0.05; * *p* < 0.05; ** *p* < 0.01; and *** *p* < 0.001).

## 5. Conclusions

Our results demonstrated that TOM40 regulates the mitochondrial activity and improves cellular energy and redox status to promote the growth of EOC. TOM40 may be an effective therapeutic target to inhibit ovarian cancer growth, although the detailed molecular mechanisms of TOM40-expression energy regulation have yet to be elucidated. Directly targeting TOM40 may be challenging in clinical application due to its substantial expression in normal cells. Therefore, as shown in our study, we propose that metformin can be a good alternative drug for targeting TOM40 and the mitochondria in epithelial ovarian cancer since metformin is already clinically used and is known to have fewer side effects.

## Figures and Tables

**Figure 1 cancers-12-01329-f001:**
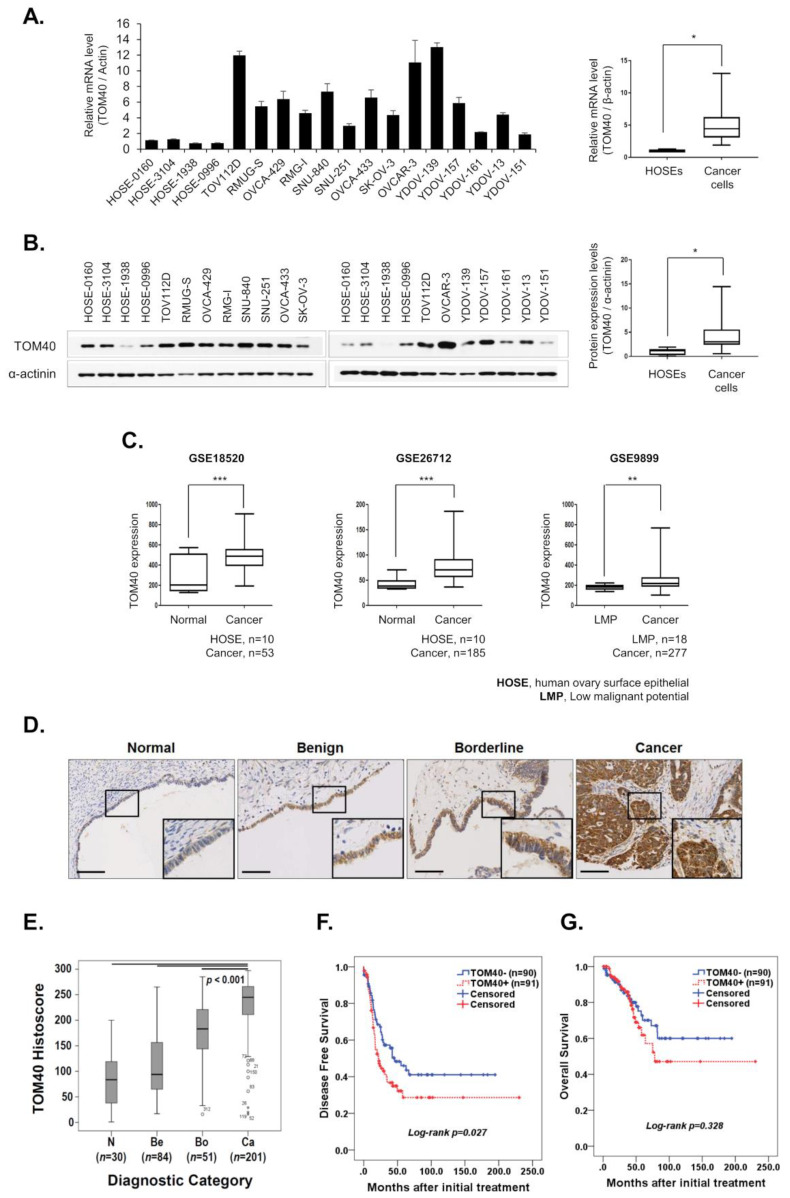
TOM40 is highly expressed by human epithelial ovarian cancer (EOC) cells. (**A**) TOM40 mRNA levels were measured by real-time polymerase chain reaction (PCR) in human ovarian surface epithelial (HOSE) cells and EOC cell lines. Fold expression is expressed as the ratio of TOM40 mRNA/actin mRNA. Results represent the mean ± Standard Error., *n* = 3. Box plots were used to compare the expression levels of TOM40 in HOSE cells and EOC cell lines; * *p* < 0.05. (**B**) TOM40 protein expression levels were examined by western blot analysis in HOSE cells and EOC cells. TOM40 band intensities were quantified relative to α-actinin using Image J 1.48v software (Right, box plot); * *p* < 0.05. (**C**) mRNA expression levels of TOM40 in the tumors of patients with ovarian cancer were analyzed using data from the Gene Expression Omnibus (GEO) database (GEO accession numbers GSE18520, GSE26712, and GSE9899). *p* values compare normal tissue or tissue with low malignant potential (LMP) to cancerous tumor tissue (** *p* < 0.01 and *** *p* < 0.001). (**D**) Representative images of immunohistochemical staining of TOM40 expression in normal epithelial ovarian tissue and in benign, borderline, and cancerous ovarian tumors: The boxed regions are displayed at high magnification in the inset (scale bar: 100 μm). (**E**) Box plot depiction of immunohistochemical staining data. The histoscores were computed based on the intensity and tissue area of positive staining. N, normal; Be, benign, Bo, borderline; Ca, epithelial ovarian cancer. (**F**,**G**) Kaplan–Meier plots of disease-free survival and overall survival for ovarian cancer patients as categorized by TOM40 expression. Survival data for only 181 of 201 patients with EOC were available to construct Kaplan–Meier survival curves. The detailed information regarding statistical tests is available in the Materials and Methods section.

**Figure 2 cancers-12-01329-f002:**
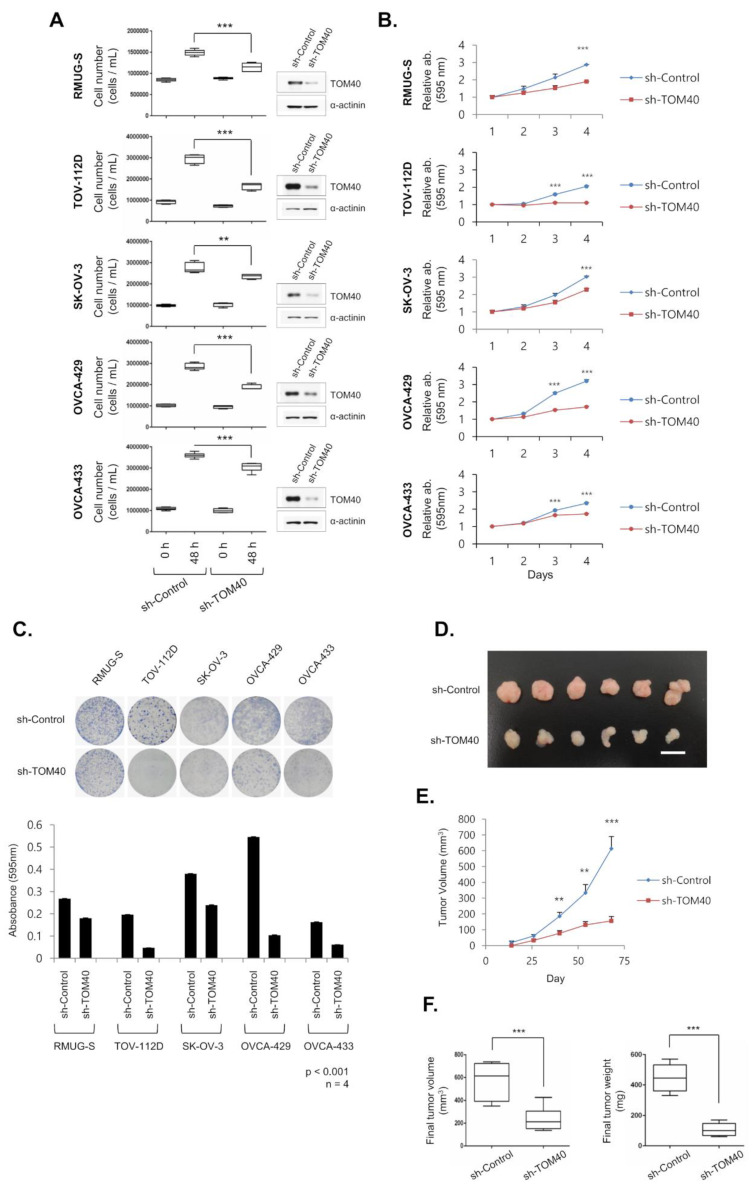
TOM40 promotes the proliferation of epithelial ovarian cancer (EOC) cell lines and EOC xenografts. EOC cell lines (RMUS-S, TOV-112D, SK-OV-3, OVCA-429, and OVCA-433) that stably express sh-control and sh-TOM40 were established by lentivirus and selected with 3 μg/ml puromycin for 15 days. (**A**) Numbers of EOC cells that stably express sh-TOM40 were counted 48 hours after seeding with a LUNAII^TM^ automated cell counter. Data are expressed as the mean ± minimum to maximum, *n* = 4; ** *p* < 0.01 and *** *p* < 0.001. TOM40 knockdown was examined by western blot analysis using an anti-TOM40 antibody (images to the right of each box plot). (**B**) The comparative growth rate of sh-control and sh-TOM40 ovarian cancer cell lines was measured daily with a crystal violet assay on days 1 through 4 post-seeding. Data are expressed as the mean ± Standard Deviation (S.D.)., *n* = 4; *** *p* < 0.001. (**C**) The long-term growth rates of EOC cells that stably express sh-TOM40 were evaluated via a colonization assay (RMUG-S, TOV-112D, and SK-OV-3 were evaluated 10 days post-seeding; OVCA-429 and OVCAR-433 were evaluated 11 days post-seeding). (Images, top half of panel) Representative images of crystal violet staining of EOC cells that stably express sh-control or sh-TOM40. (Bar graph, lower half of panel) Crystal violet intensities were measured at 595 nm using an ELISA reader. Data are expressed as the mean ± S.D., *n* = 4. (**D**) BALB/c nude mice were inoculated subcutaneously into both flanks with 3 × 10^6^ RMUG-S cells that stably express sh-control or sh-TOM40. Six tumors were used in each group. Shown is an image of the aforementioned xenograft tumors, which were dissected from the mice 68 days after inoculation. The scale bar is 5 mm. (**E**) Tumor volume was monitored at 14, 26, 40, 54, and 68 days after inoculation with RMUG-S cells that stably express sh-control or sh-TOM40. Tumor growth is expressed as the mean tumor volume ± Standard Error (S.E.), n = 6; ** *p* < 0.01 and *** *p* < 0.001. (**F**) Final volume (left box plot) and final weight (right box plot) of tumors were measured after sacrifice at 68 days post-inoculation. Data are expressed as the mean ± minimum to maximum, *n* = 6; *** *p* < 0.001.

**Figure 3 cancers-12-01329-f003:**
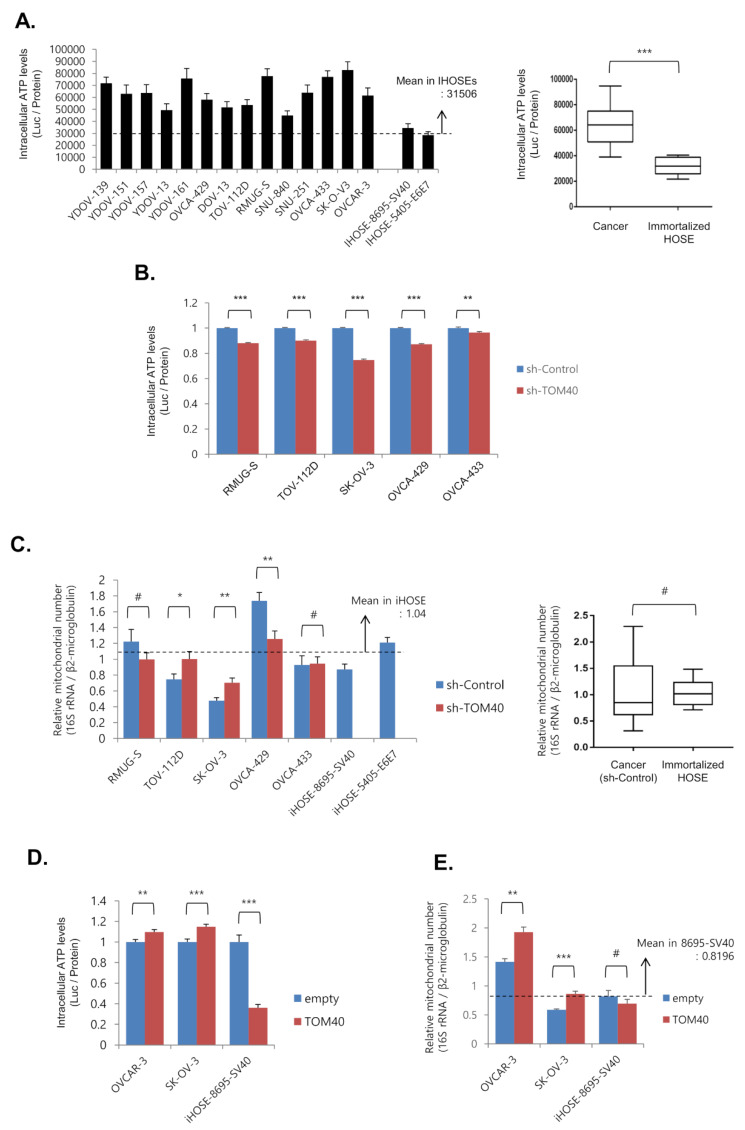
TOM40 expression is positively correlated with intracellular ATP levels. (**A**) Intracellular ATP levels were measured with a luciferase activity assay in whole cell lysates from epithelial ovarian cancer (EOC) cell lines and immortalized HOSEs (iHOSE). Luciferase activity (ATP levels) is expressed as a ratio of luciferase levels to protein quantity (bar graph, left). Results represent the means ± Standard Error (S.E.)., *n* = 4. Box plot (right) represents the comparative intracellular ATP levels of EOC cells compared to iHOSE cells. (Cancer, *n* = 14; iHOSE, *n* = 2). (**B**,**D**) Intracellular ATP levels were measured with an ATP assay kit based on luciferase activity. The relative intracellular ATP levels were quantified as a ratio of luciferase levels to protein quantity. Results are the mean ± S.E. (B, *n* = 12; *D*, cancer cell, *n* = 20; iHOSE, *n* = 8; ** *p* < 0.01 and *** *p* < 0001). (**C**,**E**) Cellular mitochondrial levels were measured by real-time PCR using total cellular DNA samples. The relative mitochondrial number was quantified as a ratio of mitochondrial 16S rRNA/ß2-microglobulin levels. Fold induction was calculated as a relative expression by iHOSE-empty vector cells. Results represent the mean ± S.E. (C, *n* = 9; E, n = 5; ^#^
*p* > 0.05, * *p* < 0.05, ** *p* < 0.01, and *** *p* < 0.001). Box plot in Figure 3C represents the relative mitochondrial number of sh-control EOC cells compared to sh-control iHOSE cells (cancer, *n* = 5; iHOSE, *n* = 2).

**Figure 4 cancers-12-01329-f004:**
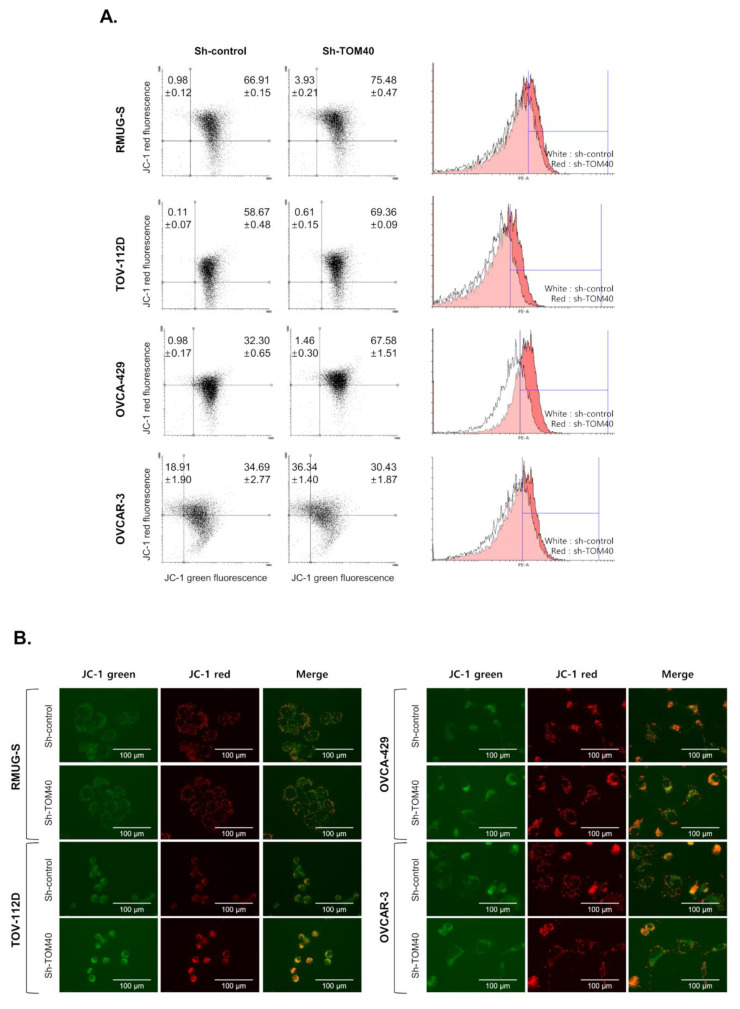
The knockdown of TOM40 expression increases the mitochondrial membrane potential. Epithelial ovarian cancer (EOC) cells that stably express sh-control and sh-TOM40 were seeded at 2.5 × 10^6^ cells per wells and stained with 2 μM tetraehylbenzimidazolylcarbocyanine iodide (JC-1) for 15 min. The JC-1 staining was measured by fluorescence-activated cell sorting (FACS) analysis and fluorescence microscopy imaging. (**A**) Representative images of FACS analysis of 5,5’,6,6’-tetrachloro-1,1’,3,3’-tetraethylbenzimidazolylcarbocyanine iodide (JC-1) staining: Quantification of the percentage of cells is shown in the top-left and top-right regions of dot plots to count JC-1 aggregation (red fluorescent). The overlay histograms are representative images of three separate experiments (The white area is sh-control cells, and the red area is sh-TOM40 cells). The results are expressed as the mean ± Standard Deviation (S.D.)., *n* = 3. (**B**) Representative fluorescent images of JC-1 staining images: The scale bar is 100 µm. EOC cells that stably express sh-control and sh-TOM40 were trypsinized, and 1 × 10^6^ cells were confined into a round-bottom tube. The cells were stained with a mixture of 100 nM MitoTracker Green FM and 25 nM tetramethylrhodamine ethyl ester (TMRE) for 15 min at 37 °C CO_2_ incubator (5% CO_2_ and 37 °C). The levels of MitoTracker Green FM and TMRE staining were measured using FACS analysis. (**C**) Representative images of histograms of MitoTracker and TMRE staining: Quantification of the percentage of cells is shown in the blue line region of histogram. The white area is sh-control cells, while the green or red area is sh-TOM40 cells in histogram. The bar graph displays the relative stained cells. The results are expressed as the mean ± S.D., *n* = 4 (TOV-112D and OVCAR-3); n = 6 (OVCA-429 and RMUG-S). ^#^
*p* > 0.05; * *p* < 0.05; *** *p* < 0.001; and **** *p* < 0.0001. (**D**) Representative fluorescent images of TMRE staining of EOC cell lines that stably express sh-control or sh-TOM40: The Scale bar is 200 μm.

**Figure 5 cancers-12-01329-f005:**
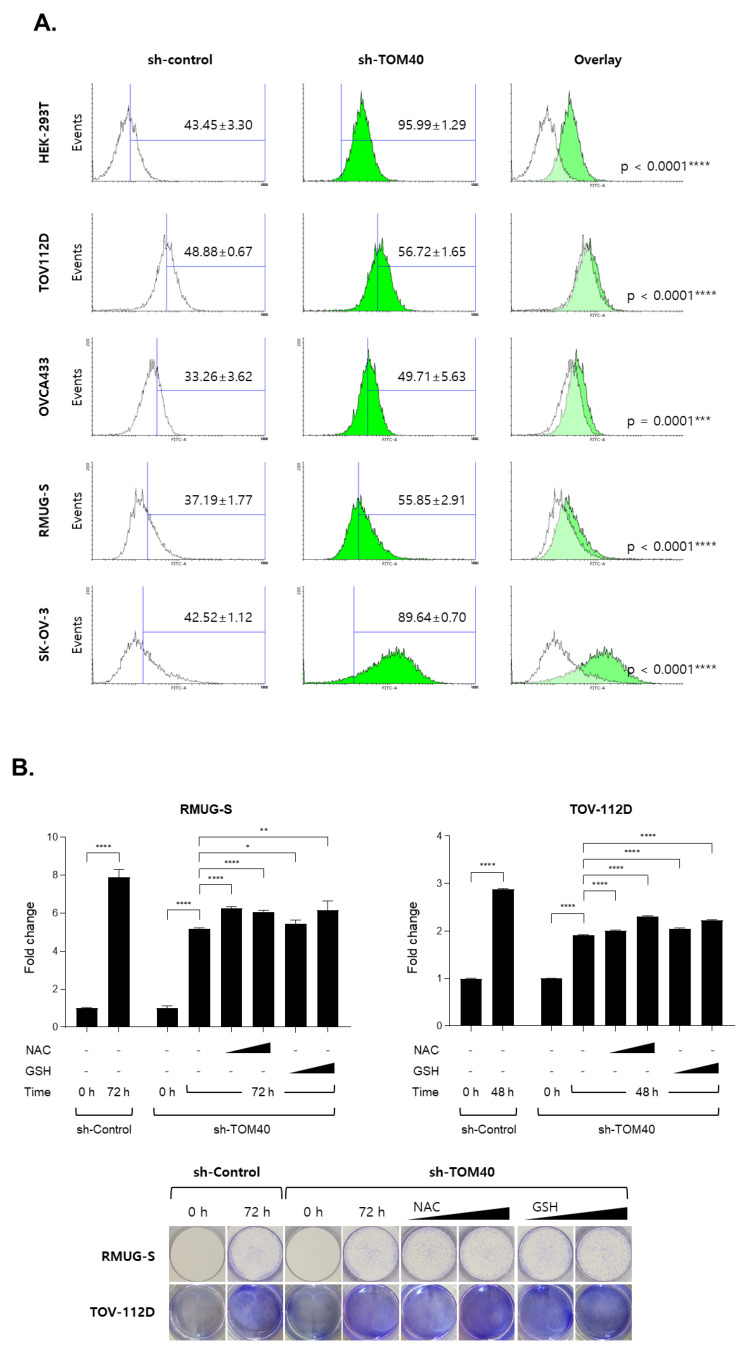
The knockdown of TOM40 expression generates intracellular reactive species which reduce cell growth. (**A**) Epithelial ovarian cancer (EOC) cells that stably express sh-control and sh-TOM40 were seeded at 1 × 10^6^ cells in a 60-cm culture dish. The cells were stained with 20 μM 2’,7’-dichlorofluorescin diacetate (DCFDA) for 30 min at CO_2_ incubator (5% CO_2_ and 37 °C). The levels of DCFDA staining were measured by FACS analysis. Representative images of histograms: quantification of the percentage of cells in shown the blue line region of histogram. The white area is sh-control cells, while the green is sh-TOM40 cells in histogram. The results are expressed as the mean ± S.D., *n* = 4 (TOV-112D and OVCAR-3); *n* = 6 (OVCA-429 and RMUG-S). *** *p* < 0.001 and **** *p* < 0.0001. (**B**) EOC cells that stably express sh-control and sh-TOM40 were seeded at 3 × 10^5^ RMUG-S and 2 × 10^5^ TOV-112D cells in a 24-well plate. The sh-TOM40-expressed RMUG-S cells were treated with 0.25 mM N-acetyl-L-cysteine (NAC), 0.5 mM NAC, 0.5 mM glutathione (GSH), or 1 mM GSH for 72 h. The sh-TOM40 expressed TOV-112D cells were treated with 0.25 mM NAC, 1 mM NAC, 0.5 mM GSH, or 1 mM GSH for 48 h. The growth rate was measured by crystal violet assay. Representative images of the crystal violet stained cells (bottom panel): Data in the bar graph are expressed as the mean of fold change ± S.D., n = 4. * *p* < 0.05; ** *p* < 0.01; and **** *p* < 0.0001.

**Figure 6 cancers-12-01329-f006:**
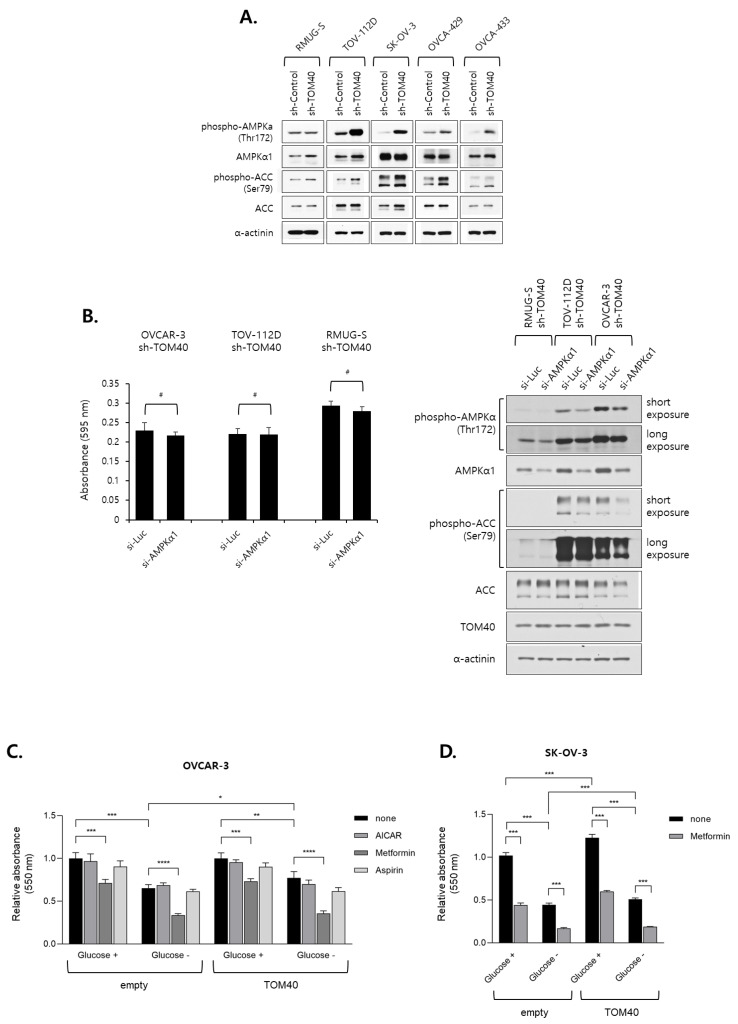
Metformin reduces EOC cell growth in an AMP-activated protein kinase (AMPK)-independent manner. (**A**) Epithelial ovarian cancer (EOC) cell lines that stably express sh-control or sh-TOM40 were harvested at 70% confluency. The expression levels of indicated proteins were measured by western blot analysis. (**B**) sh-TOM40 stably expressed EOC cells were introduced to si-Luciferase or si-AMPKα1 and then incubated in a CO_2_ incubator (5% CO_2_ and 37 ℃) for 72 h. Cell viability was assessed by crystal violet assay (Left panel). Data in the bar graph are expressed as the mean of absorbance ± Standard Deviation (S.D.)., n = 16. ^#^
*p* > 0.05. The expression levels of indicated proteins including anti-AMPKα1 were measured by western blot analysis (Right panel). (**C**) Cell viability (relative absorbance at 550 nm) as measured by 3-(4,5-dimethylthiazol-2-yl)-2,5-diphenyltetrazolum bromide (MTT) assay. OVCAR-3 that stably express an empty vector or TOM40 was cultured with or without 25 mM glucose and with or without 1 mM AICAR, 5 mM metformin, or 1 mM aspirin for 48 h. AICAR, 5-aminoimidazole-4-carboxamid ribonucleotide. (**D**) SK-OV-3 that stably express an empty vector or TOM40 was cultured with or without 25 mM glucose and with or without 5 mM metformin for 48 h. Results are shown as the relative means ± S.D., n = 6; ^#^
*p* > 0.05, * *p* < 0.05, ** *p* < 0.01, *** *p* < 0.001, and **** *p* < 0.0001. (**E**) Western blot analysis of proteins expressed by EOC cell lines, OVCAR-3 and SK-OV-3, that stably express empty vectors or TOM40 and were cultured in dulbecco modified eagle medium (DMEM) with or without 25mM glucose and with or without 1 mM AICAR or 5 mM metformin for 24 h.

**Table 1 cancers-12-01329-t001:** Clinicopathologic significance of TOM40 protein expression in epithelial ovarian cancers.

Gene	TOM40		
Parameter	No.	Mean IHC ^a^ Score (95% CI)	*p*-Value
**All study subjects**	366	184 (176–192)	
**Diagnostic Category**			***<0.001***
Normal	30	82 (63–101)	
Benign	84	112 (98–126)	
Borderline	51	176 (158–194)	
Cancer	201	231 (225–238)	
**Age**			*0.412*
≤50	93	228 (218–239)	
>50	108	234 (225–243)	
**FIGO stage ^b^**			*0.441*
I–II	57	226 (210–241)	
III–IV	124	235 (227–243)	
Recurrence	20	226 (211–241)	
**Cell type**			*0.101*
Serous	137	235 (227–243)	
Others	64	223 (210–237)	
**Tumor grade**			*0.070*
Well + Moderate	86	226 (214–238)	
Poor	100	239 (232–247)	
**Cancer type**			***0.005***
Type I tumor ^c^	81	219 (207–232)	
Type II tumor ^d^	114	241 (233–249)	
**CA125**			*0.264*
Negative	32	241 (227–254)	
Positive (>35 U/mL)	166	230 (222–238)	
**Chemosensitivity**			*0.600*
Sensitive	141	233 (225–241)	
Resistant	41	237 (227–248)	

^a^ IHC, ImmunoHistoChemistry; ^b^ FIGO, International Federation of Gynecology and Obstetrics; ^c^ Type I tumor included endometrioid, clear cell, mucinous, and transitional tumor; ^d^ type II tumor included high-grade serous carcinoma and undifferentiated tumor; protein expression was determined through analysis of an immunohistochemically stained tissue array, as described in the Materials and Methods section.

**Table 2 cancers-12-01329-t002:** Univariate and multivariate analyses of disease-free survival and overall survival in epithelial ovarian cancer patients.

Disease-Free Survival					Overall Survival				
Variables	Univariate Analysis		Multivariate Analysis		Variables	Univariate Analysis		Multivariate Analysis	
	Hazard Ratio (95%CI)	*p*-Value	Hazard Ratio (95%CI)	*p*-Value		Hazard Ratio (95%CI)	*p*-Value	Hazard Ratio (95%CI)	*p*-Value
Age (>50)	1.62 (1.07–2.45)	0.022	1.14 (0.73–1.77)	0.554	Age (>50)	2.34 (1.22–4.47)	0.01	1.76 (0.91–3.39)	0.091
FIGO stage (≥III)	6.51 (3.26–13.00)	<0.001	4.75 (2.33–9.68)	<0.001	FIGO stage (≥ III)	4.45 (1.75–11.31)	0.002	2.62 (1.01–6.81)	0.047
Cell type (serous)	2.98 (1.74–5.11)	<0.001	1.68 (0.91–3.07)	0.093	Cell type (serous)	4.08 (1.60–10.35)	0.003	2.35 (0.91–6.07)	0.077
Grade (poor)	1.98 (1.29–3.04)	0.002	1.70 (1.10–2.62)	0.017	Grade (poor)	2.27 (1.21–4.25)	0.01	2.12 (1.12–4.00)	0.021
CA125^+^ (>35 U/mL)	2.30 (1.11–4.76)	0.024	1.01 (0.46–2.24)	0.966	CA125^+^ (>35 U/mL)	1.91 (0.68–5.35)	0.216	NA	
TOM40^+a^	1.57 (1.04–2.36)	0.028	1.73 (1.12–2.67)	0.012	TOM40^+a^	1.34 (0.74–2.43)	0.33	NA	

^a^ Cutoff value of TOM40^+^ is IHC score > 245.; CI, confidence interval; FIGO, International Federation of Gynecology and Obstetrics; NA, not applicable.

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
