# Peer review of "TOM40 Inhibits Ovarian Cancer Cell Growth by Modulating Mitochondrial Function Including Intracellular ATP and ROS Levels"

_cancers, 2020, doi:10.3390/cancers12051329_

Round 1
Reviewer 1 Report
The authors have made several modifications to the manuscript. However there are still some concerns as listed below.
- The authors claim that TOM40 inhibits ovarian cancer cell growth by modulating mitochondrial function, including intracellular ATP and ROS levels but have not provided any mechanism regarding how ATP is decreased and ROS goes up.
- The overexpression of TOM40 data does not add to the paper and in fact make it more confusing. The results are variable. Overexpressing TOM40 in cell lines that already have high endogenous expression could cause secondary effects. Hence, based on the endogenous overexpression of TOM 40, the results will vary in different cell lines.
- There is no clear connection with metformin and TOM40 expression. Metformin inhibits complex I of the ETC and therefore inhibits Oxphos which will result in decrease in ATP and inhibition of cell proliferation.
- The patient data and the in vivo mouse data is significant (Fig 1 and 2). It could have been followed by in vitro experiments to measure mitochondrial stress in sh-TOM40 or corresponding control cell lines using seahorse analyzer. The authors have shown data that sh-TOM40 cell lines have decreased ATP but it does not correlate with mitochondrial number. It would be beneficial here to test the activity of ETC complexes directly using Seahorse or Oroboros.
- Activation of AMPK in sh-TOM40 cell lines might lead to increase in autophagy, decrease in fatty acid synthesis etc which could lead to inhibition of cell proliferation. Authors should test these pathways.
- Some of the flowcytometry data (fig 4) do not look very convincing.
Author Response
Comments and Suggestions for Authors
The authors have made several modifications to the manuscript. However there are still some concerns as listed below.
- The authors claim that TOM40 inhibits ovarian cancer cell growth by modulating mitochondrial function, including intracellular ATP and ROS levels but have not provided any mechanism regarding how ATP is decreased and ROS goes up.
Answer:
Although the reviewer mentioned that the result of figure 4 in question 6 was less convinced, we suggested an increase in mitochondrial membrane potential (MMP) (Figure 4A and 4C) as a cause for decreased ATP and increased ROS. In Figure 4, two different types of MMP dyes (JC-1 and TMRE) were used to measure MMP. We found that the knock-down of TOM40 expression increased MMP in both dyes. Therefore, the TOM40 knock-down increased the MMP of the ovarian cancer cells continuously. The previous reports suggested that a continuous increase in MMP leads to an increase in ROS, inhibiting mitochondria function, and reducing ATP levels (Ref. 41-44). Ovarian cancer cells have a high mitochondrial metabolic activity (Ref. 55 and Nat Med. 2011 Oct 30;17(11):1498-503), and high mitochondrial activity would have produced a significant amount of ROS (Ref. 42-44, and Cell Cycle 2012 Apr 1; 11(7): 1445–1454). ROS is produced according to an unregulated energy production process in high energy demand cells such as cancer cells. The moderate ROS level stimulates and helps the cell growth, but it is clear that the high level of ROS leads to cell death and severe cell damage (Mol Biol Rep. 2020 May 2. doi: 10.1007/s11033-020-05467-7., Nat Rev Drug Discov. 2013 Dec;12(12):931-47., Nat Rev Drug Discov. 2009 Jul;8(7):579-91.). In this study, we confirmed that the growth rate of ovarian cancer cells restored by diminishing ROS levels, which increased by TOM40 knock-down (Figure 5B). Therefore, these findings imply that the knock-down of TOM40 triggered an increase in MMP, an increase in ROS, and a decrease in ATP level as the redox balance of the cancer cell was disturbed. Hence, this reason leads to reduce the proliferation of ovarian cancer cells.
There is mention in the discussion section, from 405 to 412 lines
[Instead, we found that TOM40 knock-down consistently induced mitochondrial membrane potential and intracellular ROS levels (Fig 4 and 5). In previous reports, when the mitochondrial membrane potential is out of the normal range (dropped or enhanced) for a long duration, that decreased cell viability and impaired mitochondria [41]. High mitochondrial membrane potential produces significantly high levels of intracellular ROS [42-44]. In turn, excessive ROS generation causes mitochondrial and cellular damage [41,44]. Hence, changes in the intracellular ATP levels following TOM40 knock-down can be caused by excessive ROS generation thought the increased mitochondrial membrane potential.]
- The overexpression of TOM40 data does not add to the paper and in fact make it more confusing. The results are variable. Overexpressing TOM40 in cell lines that already have high endogenous expression could cause secondary effects. Hence, based on the endogenous overexpression of TOM 40, the results will vary in different cell lines.
Answer:
As many researchers know, overexpression and knock-down are the most straightforward approaches to know the gene function. Although the expression level of endogenous TOM40 was increased compared to normal cells, the results in Figure 3D showed the contrary evidence that the effect of changing the ATP level by TOM40 knock-down in ovarian cancer cells OVCAR-3 and SKOV-3. Instead, the iHOSE cells with less expression of TOM40 than ovarian cancer cells (Supplementary Figure S2 and S8) showed a different response than expected when TOM40 was overexpressed (Figure 3D). The result of iHOSE does not be the secondary effect of over-expression with the increased TOM40 expression. Naturally, the gene expression pattern between iHOSE and ovarian cancer is a vast difference (Ref. 58), the results in figure 3D and 3E suggests that the overexpression of TOM40 in iHOSEs with normal gene expression patterns may lead to incorrect speculation that mitochondrial activity will increase. These results indicate that TOM40 is a protein that interacts with various proteins. Ovarian cancer has these interactive factors which are not yet known, so ovarian cancer has shown that overexpression of TOM40 increases ATP levels and mitochondrial activity (Figure 3D). Although some results are occasionally different than expected, it is believed that overexpression experiments complement rather than confuse research conclusions.
It is understood that the reviewer suggested that it may be better to compare the difference between low and high cell lines, based on the expression of endogenous TOM40 than to overexpress TOM40 in ovarian cancer cells that have already overexpressed TOM40. However, it is difficult to interpret the effect of TOM40 without manipulating in the expression of TOM40 between ovarian cancer cell lines with different genetic backgrounds. As an example, in this study, while the expression level of TOM40 (Figure 1) and mitochondrial number (Figure 3E) are higher in OVCAR-3 than those in SKOV-3, the amount of intracellular ATP is lower in OVCAR-3 than in SKOV-3 (Figure 3A). Nevertheless, the growth rate of OVCAR-3 is faster than SKOV-3 (Ref. 58, Figure 1E, and Int J Hyperthermia. 2019;36(1):9-20, Figure 1A). These results are a reason why it is difficult to conclude the relationship between ATP and growth rate with only the expression level of endogenous TOM40 among ovarian cancer cells. It is also the reason why the TOM40 overexpression experiment is necessary for some experiments.
- There is no clear connection with metformin and TOM40 expression. Metformin inhibits complex I of the ETC and therefore inhibits Oxphos which will result in decrease in ATP and inhibition of cell proliferation.
Answer:
We did not suggest that metformin has a direct connection to TOM40 in this study. Ovarian cancer has high mitochondrial activity (Ref. 55 and Nat Med. 2011 Oct 30;17(11):1498-503). According to this study and previously published reports, TOM40 is closely correlated with mitochondrial metabolic activity (Ref. 13, 38-40). Based on the two possibilities, metformin has a function of reducing cell growth by inhibiting the mitochondrial activity; thus, we have suggested that metformin is a suitable drug for inhibiting ovarian cancer with high mitochondrial activity.
There is mention in the discussion section, 454 to 460 lines
[Notably, high-dose metformin suppressed mitochondrial complex I activity by inhibiting NADH oxidation [54]. Previous reports suggested that overexpressed TOM40 increased activities of the oxidative phosphorylation complexes I and IV [38], also, most ovarian cancer cell lines have higher mitochondrial activity than iHOSE cell lines [55]. Therefore, we assumed that metformin significantly reduced EOC proliferation in an AMPK-independent manner in endogenous or exogenous TOM40 overexpressed EOC cell lines.]
- The patient data and the in vivo mouse data is significant (Fig 1 and 2). It could have been followed by in vitro experiments to measure mitochondrial stress in sh-TOM40 or corresponding control cell lines using seahorse analyzer. The authors have shown data that sh-TOM40 cell lines have decreased ATP but it does not correlate with mitochondrial number. It would be beneficial here to test the activity of ETC complexes directly using Seahorse or Oroboros.
Answer:
We do not have a seahorse analyzer, so the assay using a seahorse analyzer is difficult. Thus, in order to measure the activity of ETC or Oxphos (Oxidative phosphorylation), we performed three indirect methods. The dehydrogenase activity and cytochrome c oxidase activity were measured, and the major protein expression level of ETC was detected by western blot analysis (Supplementary figure 3 to 5). In conclusion, it was not possible to conclude because the results of the TOM40 knock-down were not consistent in the cells used. Although we did not measure all the activities of five complexes of ETC, we measured the activity of complex I, II (Figure S3), IV (Figure S4), and expressed the major subunits of five complexes (Figure S5). We think it reflects some extent for ETC activity. Therefore, even if we directly measure oxygen consumption, we cannot conclude to determine the effects of TOM40 knock-down on oxygen consumption because the results will not be consistent in each cancer cell, similar to the above. It needs to proceed with the study according to the characteristics of each cell individually in order to determine the detailed mechanism of the TOM40 function. However, considering the aim of this study is the clinical significance of TOM40 expression in overall ovarian cancer patients and its role in ovarian cancer cell growth, we believe that the role of TOM40 in overall ovarian cancer has been sufficiently demonstrated in this study. Besides, as suggested by the reviewer in the previous question, it is a significantly important study to divide groups according to endogenous TOM40 expression and to study on differences about detailed mechanisms in each cell, but it is out of the scope in this study.
- Activation of AMPK in sh-TOM40 cell lines might lead to increase in autophagy, decrease in fatty acid synthesis etc which could lead to inhibition of cell proliferation. Authors should test these pathways.
Answer:
The activity of AMPK is significantly increased in sh-TOM40 cell lines compared to sh-control cells. However, the AMPK knock-down in sh-TOM40 cell lines was no change in cell proliferation rate. Besides, the effects of AICAR and aspirin, known as activators of AMPK, were minimal in cell proliferation. Only metformin, which has the function of inhibiting the activity of mitochondria, inhibited the proliferation of ovarian cancer, so we found that AMPK is not correlated with cell proliferation by TOM40 knock-down. Even if AMPK is related to autophagy or fatty acid synthesis, it does not need to check the AMPK-related signals at this point.
Beta-oxidation may increase by TOM40 knock-down. It can be expected in the inhibition of ACC activity due to the phosphorylation of ACC (Figure 6A and 6B), which is involved in fatty acid synthesis. The previous report showed that ovarian cancer is frequently metastasized to the omentum. Metastasized ovarian cancer uptakes fatty acid from adipocyte in the omentum and uses it as an energy source. Then, ovarian cancer gets to have more malignant features (Nat Med. 2011 Oct 30; 17 (11): 1498-503). However, this phenomenon does not occur at the monolayer culture level, and it happened only in a co-culture with an adipocyte or in vivo mouse. As shown in a mouse experiment in this study (Figure 2D to 2F), the inhibition of ovarian cancer growth by TOM40 knock-down showed that cancer does not gain an advantage for growth even if the energy source was converted to beta-oxidation for energy because the overall mitochondria activity was inhibited.
Lastly, after TOM40 knock-down, an increase in LC3, a key regulator of autophagy, was observed in some ovarian cancer cells. This phenomenon does not find in all the ovarian cancer cells used in this study, and like the described in the manuscript, our results showed differences between ovarian cancer cells. We hypothesized that the TOM40 knock-down could be reduced the mitochondrial number by mitophagy, which led to that ATP levels decrease in part of ovarian cancer cells. The autophagy-related results were not disclosed because this project was newly started.
- Some of the flowcytometry data (fig 4) do not look very convincing.
Answer:
Figure 4 shows the result of increased mitochondrial membrane potential by sh-TOM40 knock-down when staining with JC-1 and TMRE. Since there is no description of why figure 4 is not convincing, we are not sure how to respond to this comment. If the reviewer is asking a question to reinforce the reviewer’s thoughts on the first comment, then the answer to the first question will be sufficient. If the mitotracker green staining in Figure 4C is confused with the interpretation, the mitotracker results can be transferred to the supplementary section or deleted, as the mitotracker does not affect the MMP increase.

Reviewer 2 Report
No comments for the authors
Author Response
Thank you for your comments and suggestions which have significantly helped us to improve our manuscript.
Round 2
Reviewer 1 Report
Authors have addressed all concerns.
This manuscript is a resubmission of an earlier submission. The following is a list of the peer review reports and author responses from that submission.
Round 1
Reviewer 1 Report
The manuscript by Yang et al investigates the role of TOM40, a mitochondrial protein translocase, in ovarian cancer. The manuscripts is based on the initial observation that TOM40 is overexpressed in ovarian tumors and in ovarian cancer cell lines. The authors then investigate the effect of TOM40 overexpression on expression of other mitochondrial proteins as well as on ATP production. Finally, the authors correlate the effect of TOM40 overexpression on AMPK activation. There is considerable experimental data provided in the manuscript and the observations are interesting and potentially have high significance in understanding mitochondrial function in ovarian tumors. However, there are also several shortcomings that reduce enthusiasm for this study.
1. The manuscript is arranged in a confusing manner. For example, even in the introduction, the paragraph on AMPK is not well integrated with the rest of the information provided on TOM40.
2. The major question that the authors should have addressed is what is the role of TOM4 expression in ovarian cancer. However, the studies do not provide a clear understanding on TOM40. The experiments conducted are not focused and jump around from one to another without providing a clear understanding. This is unfortunate because there is so much potential in these studies.
3. The studies with AMPK and metformin are not conclusive in showing the role of TOM40.
4. The experiments conducted with cell lines overexpressing TOM40 are not fully justified. On one hand, there is data provided that ovarian cancer cells lines are overexpressing TOM40. Therefore, making these cell lines express even more TOM40 is likely not going to produce data that can be properly interpreted.
5. It will be more informative if the authors have conducted experiments to directly determine the effect of TOM40 overexpression on OXPHOS activity. There is already information in the manuscript that TOM40 knockdown results in decrease of SDHA and other proteins. Therefore, a follow up study to monitor OXPHOS activity (using Seahorse analyzer, for example) would have been more informative.
6. The experiments with metformin are also complicated by the fact that this drug is also known to inhibit OXPHOS. Therefore, the interpretation of the results is not straightforward.
7. An increase in ROS could be a sign of improper OXPHOS pathway. This is an observation that should have been followed with more direct assays.
8. To a great extent authors are missing the point that TOM40 overexpression could be a sign that the ovarian tumors are attempting to normalize or even expand their reliance on OXPHOS. This is definitely an observation that should be further investigated.
9. Finally, there are various typographical and grammatical errors throughout the manuscript that should be corrected.
Reviewer 2 Report
Author here described the functional role of TOM40 in ovarian cancer whereby they showed the effects are due to AMPK activation.
TOM complex are always linked to TIM complex for mitochondrial protein import which has been totally ignored in the current study. Expression of TIM complex proteins after TOM40 knock down or over expression should be checked. Did authors found any correlation between TOM and TIM complex proteins in Geodatasets. Authors should also check the quality of mitochondria in TOM40 knock down or overexpression system which might correlate with intracellular ATP levels as opposed to mitochondrial quantity. It will be interesting to look at the mitochondrial morphology (fission or fusion state) by fluorescent microscopy after TOM40 knock down or overexpression. One of the main conclusion from the authors is that TOM40 effects are dependent on AMPK activation, however, data from Figure 6 suggests that effects were independent of AMPK. Both AMPK activation as well as inhibition showed the EOC cell growth inhibitory effects questioning the overall conclusion of the study. Authors should check the alternative cell survival pathways.